# Brief communication: Post-seismic landslides, the tough lesson of a catastrophe

Xuanmei Fan[1], Qiang Xu[1], Gianvito Scaringi[1]

[1] The State Key Laboratory of Geohazards Prevention and Geoenvironment Protection (SKLGP), Chengdu University of Technology, Chengdu, Sichuan, China, 610059.

Correspondence to: Qiang Xu, xuqiang_68@126.com and Gianvito Scaringi, g.scaringi@qq.com

**Abstract**

The rock avalanche that destroyed the village of Xinmo in Sichuan, China, on June 24th, 2017, brought the issue of landslide risk and disaster chain management in highly seismic regions back into the spotlight. The long-term post-seismic behaviour of mountain slopes is complex and hardly predictable. Nevertheless, the integrated use of field monitoring, remote sensing and real-time predictive modelling can help to set-up effective early warning systems, provide timely alarms, optimize rescue operations and perform secondary hazard assessments. We believe that a comprehensive discussion on post-seismic slope stability and on its implications for policy makers can no longer be postponed.

## 1 Introduction: the 2017 Xinmo landslide, the lasting legacy of earthquakes

On June 24th, 2017, after days of not-so-heavy rain, a 13 million m$^3$ rock and debris avalanche submerged the village of Xinmo (in the eastern margin of the Tibetan plateau, Sichuan, China) with impressive energy, rushing towards the river and blocking its course for more than 1 kilometre. The rescue operations were launched promptly, and all possible efforts were done by local heroes and professionals. Nonetheless, 10 people were found dead and further 73 were reported missing in one of the deadliest landslides in recent history (Fan et al., 2017; Figure 1).

Almost a century earlier, in 1933, Xinmo was struck by a magnitude M$_s$ 7.5 earthquake, during which large-scale landslides destroyed villages and choke rivers, producing dammed lakes that, collapsing, produced enormous floods, killing thousands of people (Cheng et al., 2008). Many coseismic landslides left large amounts of loose material along the slopes which, in turn, caused

deadly debris flows and avalanches during every rainy season, for decades. Just like it happened, and it is still happening since 2008, after the $M_s$ 8.0 Wenchuan earthquake (Fan and Huang, 2013), not many kilometres apart. However, what does not collapse during the quake is not exempt from damage. An earthquake can produce cracks and fractures in the rock which, paradoxically, can be noticed from space by the eyes of satellites, but be hidden to the human eye if covered by dense vegetation on high-elevation ridges (e.g. Fan et al., 2017). The damaged rock can hold in place for decades or even centuries, but rainwater can infiltrate within the fractures, dissolve minerals, fill the cracks, freeze, pull the blocks apart with its pressure until, more or less suddenly, the proverbial last straw – a rainfall or a minor shake (Qiu, 2016) – will make it collapse.

The long-term effect of strong earthquakes on the geological hazards in mountainous areas seems to be an underestimated issue. While coseismic landslides are well-described (e.g. Parker et al., 2011; Zhang et al., 2016), and large attention is being given to the short and mid-term effects of earthquakes on debris flow occurrence and sediment yield (Hovius et al., 2011), the delayed effects on slope stability is often neglected. Rock weathering and crack propagation are complex time-dependent processes. Thus, the occurrence of post-seismic landslides does not follow a clear trend, and destructive events might happen "randomly", decades after the quake (see Towhata, 2013). Rock slopes damaged by the $M_w$ 7.7 Chi-Chi earthquake in 1999 in Taiwan (Lin et al., 2006) are still collapsing, year after year during the rainy seasons, causing hundreds of fatalities. Post-seismic landslides and long-term rock degradation have been reported in several areas of Japan (Okamoto et al., 2012), and a clear dependency of landslide occurrence with past earthquakes has been found in New Zealand (Parker et al., 2015). Several recent landslides in Sichuan, China, have been correlated to the $M_s$ 7.5 Xichang earthquake (Wei et al., 2014), which occurred more than 160 years ago.

Five to ten thousand people die because of a non-coseismic landslide each year in the world, with China being the most exposed country (Petley, 2012) and with a worrisome increasing trend over time. A significant number of events can be regarded as post-seismic landslides, such as the recent one in Xinmo, for which the rock weakening action of past strong earthquakes likely acted as a predisposing factor.

## 2 An example of policy: disaster prevention and emergency response in China

China has been doing great efforts in funding research, allocating special budget to professional teams and training the general public and government divisions at various levels with the aim of improving the early reconnaissance, warning and prevention of geological hazards. Nevertheless, tragedies still happen and, unfortunately, the Xinmo landslide was not an isolated event. In 2013, the Wulipo landslide – an event with similar characteristics – made 166 victims. It was considered as a lagged effect of the 2008 Wenchuan earthquake (Yin et al., 2016). According to Huang (2009), most of the catastrophic landslides in China might be caused by the joint effect of earthquakes and rainfall (Huang, 2009).

On the other hand, China has gained solid experience in the emergency response and rescue in catastrophic events. With the *leitmotif* "safety first", the government listed the disaster prevention and reduction in its economic and social development plan as an important guarantee of sustainable development. One year after the 2008 Wenchuan earthquake, with the white paper "China's Actions for Disaster Prevention and Reduction", the State Council defined the strategic goals and tasks of disaster reduction and built a legal framework, an institutional setup and an operative mechanism for disaster reduction (Chinese Government, 2009). To strengthen the capacity of emergency rescue and relief work, the National Emergency Plan for Sudden Geological Disasters was also enforced, featuring centralized command, sound coordination, clear division of tasks and level-by-level control with local authorities. The geo-disasters have been classified into four size-dependent categories according to the estimated fatalities and money loss (Figure 2). Different levels of government have been given responsibility for handling disasters of different magnitude.

In the wake of a highly catastrophic event, the local-level divisions are required to report to the State Council directly within no more than 4 hours, and this latter shall take immediate action. The commanding and coordination headquarter, led by the Council directly, shall be set up with a cross functional steering committee, consisting of experts from different fields, to conduct rescue, evacuation, temporary relocation, information and data gathering, geological survey, weather forecast, medical and epidemic prevention, lifeline engineering repair, and so on. The headquarter has also the power to command the People's Armed Police (the Chinese army) directly. Conversely, in case of small and medium-scale disasters, the local government shall trigger the

emergency response immediately and autonomously, and set up a local emergency command, with the local government's heads serving as chief commanders, to jointly set up the emergency response and disaster relief, organize the field work and report on the disaster details and work progress to the higher governmental level.

**3 Open discussion: the lesson of a killer landslide**

The recent catastrophic event in Xinmo received considerable attention by the scientific community and by the media worldwide, bringing the issue of landslide risk and disaster chain management in highly seismic regions back into the spotlight. Through this brief, work-in-progress paper, we hope to trigger a comprehensive open discussion within the scientific community and gather and share ideas on the best handling of long-term post-seismic slope stability problems and on its implications for policy makers.

Soon after the Xinmo landslide, the Sichuan Province Administration began an inch-by-inch investigation to identify potential geohazards before the beginning of the rainy season. The satellite radar interferometry technique (InSAR) has been applied to identify hot spots of deformation within the large search region. Laser scanning (LiDAR) and drone flights (UAV) have then been used to further confirm the potential hazardous sites. We think that this is perhaps the most effective way to proceed in densely vegetated mountainous areas. Satellite images of the Xinmo village in the visible spectrum taken since 2003 showed, indeed, several cracks, up to 150 m long and interconnected to some extent, in the landslide source area (Fan et al., 2017). InSAR images highlighted noticeable deformations (Figure 3a, b) in the rock mass during the months preceding the landslide and, reasonably, the infiltration of recent rainfalls within the cracks speeded up the failure process (Fan et al., 2017).

Tragedies such as the one occurred in Xinmo might be avoided if the same scrupulous and systematic early reconnaissance and monitoring activity is carried out in due time. Satellite imagery can help detect and prioritize potential hazardous areas. Then, through field and aerial investigations, using UAV, LiDAR and InSAR, a detailed mapping can be done. Potentially critical situations can be recognized and then handled through continuous monitoring of deformations, for instance through ground-based SAR interferometry (Crosetto et al., 2014; Monserrat et al., 2014; Fan et al., 2017). Furthermore, in-situ recording and interpretation of the

characteristics of ambient noise, i.e. the low energy and low frequency vibrations of the ground due to natural or anthropic sources (e.g. Del Gaudio, 2017), can be used to characterise the mechanical properties of the soil or rock mass (e.g. anisotropy, sets of directional discontinuities; Di Giulio et al., 2009) and their changes with time (e.g. shear wave velocity decrease due to crack opening or water level rise; Daskalakis et al., 2016; Behm, 2016), potentially providing an indirect assessment of strength degradation or stress changes. Furthermore, monitoring of acoustic emissions, i.e. the high frequency transient elastic waves originating from the sudden release of energy at localized points within a loaded material (Nomikos et al., 2010), can provide information on the occurrence of microcracks within the rock, that may signal progression of rock damaging, crack coalescence, deformations and incipient failure (e.g. Manthei and Eisenblätter, 2008; Agioutantis et al., 2016). Such techniques, combined with hydrological monitoring, would be extremely precious to set-up early warning systems and help the authorities to take informed decisions on possible evacuation or relocation of the exposed people.

And, if the event happens too suddenly to take countermeasures, something could still be done to optimize the alarm-and-rescue chain, for instance, by using the existing seismic networks. In fact, it is known that landslides generate seismic signals, "landquakes", which contain a specific signature: low-frequency waves released by the bedrock when the mass detaches, and high-frequency waves produced by the landslide mass while it is sliding, peaking when it impacts the deposition area (e.g. Yamada et al., 2012; Chen et al., 2013). If analysed separately, they can give information on both landslide initiation and impact (Figure 3c-g). In theory, two distinct epicentres can be identified automatically by the seismic networks, if they are sensitive enough and they are taught to do so. In the case of the Xinmo landslide, the seismic recording showed that just one minute elapsed from the initiation to the deposition, during which the mass slid along the slope for more than 2.5 km, with more than 1000 m of height relief, and hit the population at an impressive velocity $v = 250$ km/h (Fan et al., 2017, following Lin, 2015). The potential energy released by the event could be estimated, $E = 290$ TJ (Fan et al., 2017, following Lin et al., 2015), and so the volume involved, $V = 13$ million m$^3$ (pretty close to the estimation based on field observation and topographic difference), and the average mobilized friction coefficient, $\mu = 0.29$ (following Lin, 2015). For comparison, similar evaluations resulted in $E = 150$ TJ, $\mu = 0.12$, $v = 298$ km/h, and $V = 10^7$ m$^3$ for the 2009 Hsiaolin landslide in Taiwan (Lin et al., 2010; 2015; Lin, 2015), and $E = 55$ TJ, $v = 101$ km/h and $V = 2.1 \cdot 10^7$ m$^3$ for the 2011 Atakani landslide in Japan (Yamada et al., 2013).

Systematic evaluations of landslide characteristics based on seismic recordings are also given by Chen et al. (2013).

If all these calculations were done automatically, within minutes from the event, the authorities would have received a detailed alarm report containing the coordinates and magnitude of the landslide, the runout, the rock/soil involved, the volume and impact velocity, the number of people and infrastructures potentially affected and the estimated damage. Such quantitative information can be extremely useful to launch the rescue operations in the most efficient way. Some work has been done on this path already (Chao et al., 2017), and seems very promising.

Finally, after the landslide event, an accurate and continuous secondary hazard assessment is fundamental. Fan et al. (2017) reported on a preliminary evaluation of the secondary hazard deriving from potential further failures in the source area and its surroundings. Various potentially unstable masses have been identified. Among them, a large-scale deformation of a 4.5 million $m^3$ mass was detected through the interpretation of UAV images. The mass was likely displaced by the shearing and dragging action of the Xinmo landslide, but it stabilized after sliding for about 40 m after encountering a natural obstacle. During the emergency rescue operations, the mass was believed to be in a state of incipient failure and received considerable attention. In order to provide a reliable evaluation of its stability condition, and to ensure the safety of people near the landslide deposition area and preventing further disasters, a ground-based SAR was installed. Subsequently, numerical modelling with various methods (finite elements, discrete elements) was carried out to evaluate the potentially affected areas in case of a new failure (Scaringi et al., 2017). The model results showed that the potential new landslide would likely affect several more buildings and a further portion of river and of road infrastructure. Furthermore, the resulting river damming would pose a serious risk for the population living downstream in case of dam breach and for the population living upstream for the possible water level rise. In the wake of these results, different modelling approaches have been discussed comparatively, and the opportunity of an integrated real-time monitoring-and-modelling system arose. As pointed out by Molinari et al. (2014), a physically-based numerical model capable of re-computing a new solution in a very short time (i.e. within seconds) based on spatially distributed real-time field monitoring data can be extremely useful in dynamic risk assessment systems at a scale of detail to provide early-warning to the authorities and implement timely risk mitigation countermeasures.

**4 Concluding remarks**

The Xinmo tragedy taught us a tough lesson, but also showed us how the use of new technologies and the collaborative work of experts and professionals can prove successful in identifying potential hazards and performing quick assessments in "inaccessible" areas. As the hazard chain of earthquakes can last for centuries, we argue that a dedicated hazard prevention and mitigation department should be established in every region where strong earthquakes can strike. It would provide the necessary coordination and integration of resources, information, equipment and manpower. It would be able to set up big data centres and platforms, automatic reconnaissance, warning and alarm algorithms. It would carry out comprehensive research on geological hazard prevention and promote the practical application of new technologies. Finally, it would comprehensively enhance our capacity of preventing and mitigating geological hazards, and avoid tragedies.

**Acknowledgments**

After the occurrence of the June 24th Xinmo landslide, the State Council, the Ministry of Land and Resources, and various governmental departments at all levels in Sichuan Province immediately devoted their efforts into the emergency rescue operations, secondary hazard relief and geological surveying and monitoring. To them goes our most sincere appreciation and gratitude.

We thank the Sichuan Provincial Surveying and Mapping Geographic Information Bureau, the High-resolution Earth Observation System Sichuan Data and Application Center, the Sichuan Shu Tong Geotechnical Engineering Company, the Beijing Digital Greenfield Technology Co. Ltd. We express our gratitude to Prof. Yueping Yin from China Geological Environment Monitoring Institute, to Prof. Xiao Li from China Geological Survey, to Prof. Zhenhong Li from Newcastle University, to Prof. Lu Zhang and Prof. Mingsheng Liao from Wuhan University, to Prof. Qin Zhang from Chang'an University, to Prof. Shizhong Hong from Chengdu Earthquake Prevention and Hazard Mitigation Bureau, to Dr. Xinghui Huang from China Earthquake Networks Center, to Prof. Yong Li from Chengdu University of Technology, to Prof. Chong Xu from the Institute

of Geology, China Earthquake Administration, for providing valuable information at the earliest time. We express sincere thanks to Dr. Yanan Jiang, Dr. Jing Ran, Xianxuan Xiao, Weiwei Zhan, Jing Ren, Yuanzhen Ju, Chen Guo, and other postgraduate students from Chengdu University of Technology for their hard work on the UAV aerial photography, ground-based SAR monitoring and other field work.

This research is financially supported by National Science Fund for Outstanding Young Scholars of China (Grant No. 41622206), the Funds for Creative Research Groups of China (Grant No. 41521002), National Science Fund for Distinguished Young Scholars of China (Grant No. 41225011), the Fok Ying-Tong Education Foundation for Young Teachers in the Higher Education Institutions of China (Grant no. 151018), the AXA fund.the Fund from Land and Resources Department of Sichuan Province (Grant No. KJ-2015-01).

The authors are grateful to the editor, Dr. K.-T. Chan, for handling the manuscript, and to Dr. W.-A. Chao and one anonymous referee for the useful comments and for encouraging the publication of this work.

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

**Figure captions**

**Figure 1** Panoramic view of the Xinmo rock avalanche as seen from the opposite slope. Notice the red trucks for the scale

**Figure 2** The emergency response to disasters in the People's Republic of China

**Figure 3** (a, b) Differential interferogram of the landslide obtained by processing Sentinel-1 satellite data. The arrow points to the deformation occurred in the landslide source area. (c-g) Seismic signal of the Xinmo landslide event recorded by the Maoxian MXI station, about 43 km apart (modified from Fan et al., 2017): vertical component (c); frequency-time Hilbert spectrum (d); high-frequency time-magnitude spectrum, f = 2.73 Hz (e); low-frequency time-magnitude spectrum, f = 0.4 Hz (f); frequency-magnitude spectrum (g)