# Peer review of "Brief communication: Post-seismic landslides, the tough lesson of a catastrophe"

_Natural Hazards and Earth System Sciences, 2017_

## Referee Comment (RC1) · W.-A. Chao (Referee) · 21 Nov 2017

The manuscript by Xu et al., in the case of 2017 Xinmo landslide, aims to layout the logistics of how such a dynamic early warning system is possible and should be established in the region with strong seismicity. I think that the subject is relevant to publication in NHESS, especially for format of "Brief communication", but there are several places where I think a bit more explanation and minor revision are needed. More detailed comments are listed below.

Lines 100-111: "tens of meters of interconnected cracks in the landslide area", please specify the size of this precursor cracks and don't simply refer tens of meters.

Lines 126-127: "for instance through ground-based SAR, ambient noise recordings
and acoustic sensors", please add the references and paragraph on the description of "ambient noise recordings" and "acoustic sensors" for non-specialists.

Lines 134-136: Chen et al. (2013) also presented the characteristics of high-frequency seismic signals related to the different mass movements (e.g., rockfall, rock slide). Please also add a reference of Chen et al. (2013).

Chen, C. H., W. A. Chao, Y. M. Wu, L. Zhao, Y. G. Chen, W. Y. Ho, T. L. Lin, K. H. Kuo and R. M. Zhang (2013) A Seismological Study of Landquakes Using a Real-Time Broadband Seismic Network. Geophys. J. Int., 194, 885-898.

Line 140: typo error "thee".

Lines 140-144: Please replace "energy released" by "potential energy released". Did you compute aforementioned values (runout distance, drop height, sliding velocity, energy release and collapse volume) by yourself? If not, you should add the references and/or the mathematic expressions to clarify above parameters, which relates to source kinematics. You show the potential energy released during the landslide to be 290 TJ. Do you think this is a realistic value for landslides? Please also compare your results with published studies. The reader may want to find explored by the authors.

Lines 143-145: ". . .within seconds from. . .". In fact, the computing time depends mainly on the length of seismic waveforms used in the source determination. In a case of seismic waveform inversion (long-period seismic signals), a few minutes (> 100 sec) of data length is needed for an inversion scheme. Please replace "seconds" by "a few minutes".

---

## Referee Comment (RC2) · Anonymous Referee #2 · 28 Nov 2017

The short communication submitted by Xu et al. tackles the theme of landslides that may affect mountain areas struck by strong earthquakes in the past to mitigate the risk associated with them. The authors underline the need to analyze the post-seismic stability conditions of slopes using all available ground and aerial methodologies along with the use of appropriate computer models. They also highlight the need for an appropriate monitoring of the unstable slopes and an efficient management of the post-collapse emergency by the territorial authorities. The text is well organized and correctly written and could be accepted as it is.

---

## Author Comment (AC1) · 13 Dec 2017

Dear Editor and referees,

We thank you for your comments. Please find attached the supplement containing the following:

1. the detailed response to the referees' comments; 2. the revised version of the manuscript with marked changes; 3. the revised version of the manuscript with the changes incorporated.

Best regards,

Dr. Gianvito Scaringi, on behalf of my coauthors

[Figure]

Please also note the supplement to this comment:
https://www.nat-hazards-earth-syst-sci-discuss.net/nhess-2017-363/nhess-2017-363-AC1-supplement.zip